# Subversion of Host Innate Immunity by Human Papillomavirus Oncoproteins

**DOI:** 10.3390/pathogens9040292

**Published:** 2020-04-17

**Authors:** Irene Lo Cigno, Federica Calati, Silvia Albertini, Marisa Gariglio

**Affiliations:** Molecular Virology Unit, Department of Translational Medicine, Medical School, University of Piemonte Orientale, 28100 Novara, Italy; irene.locigno@med.uniupo.it (I.L.C.); federica.calati@med.uniupo.it (F.C.); silvia.albertini@med.uniupo.it (S.A.)

**Keywords:** human papillomavirus, HPV, innate immunity, E6 and E7 oncoproteins, HPV-driven cancer, pathogen recognition receptors, PRR

## Abstract

The growth of human papillomavirus (HPV)-transformed cells depends on the ability of the viral oncoproteins E6 and E7, especially those from high-risk HPV16/18, to manipulate the signaling pathways involved in cell proliferation, cell death, and innate immunity. Emerging evidence indicates that E6/E7 inhibition reactivates the host innate immune response, reversing what until then was an unresponsive cellular state suitable for viral persistence and tumorigenesis. Given that the disruption of distinct mechanisms of immune evasion is an attractive strategy for cancer therapy, the race is on to gain a better understanding of E6/E7-induced immune escape and cancer progression. Here, we review recent literature on the interplay between E6/E7 and the innate immune signaling pathways cGAS/STING/TBK1, RIG-I/MAVS/TBK1, and Toll-like receptors (TLRs). The overall emerging picture is that E6 and E7 have evolved broad-spectrum mechanisms allowing for the simultaneous depletion of multiple rather than single innate immunity effectors. The cGAS/STING/TBK1 pathway appears to be the most heavily impacted, whereas the RIG-I/MAVS/TBK1, still partially functional in HPV-transformed cells, can be activated by the powerful RIG-I agonist M8, triggering the massive production of type I and III interferons (IFNs), which potentiates chemotherapy-mediated cell killing. Overall, the identification of novel therapeutic targets to restore the innate immune response in HPV-transformed cells could transform the way HPV-associated cancers are treated.

## 1. Introduction

Human papillomaviruses (HPVs) are small, non-enveloped double-stranded DNA viruses responsible for the development of cancers of the anogenital and upper aerodigestive tract, with an incidence of ~5% among all cancers worldwide. The HPV genome is a circular double-stranded DNA episome containing one regulatory region and two early (E) and late (L) open reading frames (ORFs). Among early proteins, E6 and E7, the only two viral genes consistently found in cervical tumors, are required for the development of HPV-associated cancer. The transforming activity of these two oncoproteins is mediated primarily through protein–protein interactions conducive to a replication-competent environment that also includes the dampening of the innate immune response.

Although anogenital HPV infections affect ~80% of the population during their lifetime, they are cleared for the most part by the host immune system through a process that can take up to 1–2 years from the initial infection, with just a minority of infected individuals developing HPV-associated tumors. The deregulated expression of the two viral oncoproteins E6/E7 is necessary for tumor development and maintenance. A plethora of studies have extensively demonstrated the molecular mechanisms through which E6/E7 facilitate the accumulation of genetic mutations and chromosome abnormalities leading to cancer development, as reviewed elsewhere [1,2,3,4,5,6,7,8,9,10,11,12]. Conversely, even though escape from innate immune surveillance appears to be the hallmark of persistent HPV infection, our knowledge on the interplay between E6/E7 oncoproteins and the mediators of innate immunity in keratinocytes is still limited and has just started to be unraveled.

In light of the above, this review will provide an overview of recent studies bridging E6/E7-mediated inhibition of innate antiviral immunity and cancer progression. Understanding how the deregulated expression of E6/E7 can efficiently subvert innate and adaptive immunity may pave the way for the development of novel anticancer strategies aimed at restoring cellular reactivity in HPV-transformed cells. In this review, we summarize a growing body of literature highlighting the biological relevance of E6/E7 as suppressors of the innate immune response in HPV-driven tumorigenesis.

## 2. HPV Infection and Human Carcinogenesis

HPVs are a heterogeneous group of non-enveloped DNA viruses responsible for various types of skin and mucous membrane lesions, which generally heal without requiring any medical intervention despite retaining the potential to evolve into invasive cancers under certain conditions. HPVs have a circular double-stranded DNA genome of ~8 kb that includes non-structural E and structural L genes. To date, over 200 different types of HPV have been identified and classified into several phylogenetic groups [13]. Of these, mucosal HPVs belonging to the α-genus associated with infections of mucosal epithelia are the best characterized. They can be grouped into ‘low-risk’ and ‘high-risk’ types, depending on the relative propensity of the resulting neoplasms to undergo malignant progression, as reviewed elsewhere [3,4,14,15,16,17].

Although a dozen HPV types belonging to the mucosa-infecting alpha genus have been classified by the WHO as carcinogenic, the two most commonly found HPV types in cervical cancer are HPV16 (~50%) and HPV18 (~15%), which alone accounts for over 500,000 new cancer cases and over 250,000 cancer deaths per year worldwide. In addition, a significantly increasing prevalence of HPV-driven oropharyngeal cancers (OPSCCs) has been observed over the past few decades in western countries, with HPV16 being the most frequently found genotype. According to several studies, the impact of HPV infection on oropharyngeal cancer development is projected to exceed that of environmental factors in these regions [1,16,17,18,19,20].

In principle, HPV-driven cancer could be prevented by vaccination against oncogenic HPV types. There are in fact three currently available prophylactic vaccines on the market that efficiently protect against infection with the most common oncogenic HPV types. However, despite this encouraging outlook, there are still important issues to be dealt with, such as vaccine hesitancy—only a small minority (7.5%) of females worldwide, aged 10–20 years, are estimated to have received at least one shot of an HPV vaccine [8,15,17]. In addition, given that HPV-driven carcinogenesis is the result of persistent infection with oncogenic HPV types, often lasting several decades, it is highly likely that HPV-associated tumors will remain a major health concern for the foreseeable future, thus requiring novel effective therapeutic solutions [21,22,23,24,25,26]. In this context, feasible therapeutic options may include the restoration of the innate immune response in HPV-transformed cells. Thus, more efforts should be put in place to mechanistically dissect the cross talk between epithelial innate immunity and E6/E7 proteins in order to unveil potential therapeutic targets.

The transforming activity of E6 and E7 oncoproteins is mediated primarily through protein–protein interactions conducive to a replication-competent environment eventually leading to cancer. A plethora of interactome analyses of high-risk (hr) genotypes have identified hundreds of cellular proteins potentially interacting with E6 and/or E7, probably reflecting the large number of biological functions exerted by these two viral oncoproteins. Obviously, the most relevant biological activity of these oncoproteins is inactivation of the tumor suppressors p53 and pRb. In particular, E6 mediates the proteasomal degradation of p53, thereby disrupting its ability to induce cell cycle arrest and apoptosis in stressed cells. Similarly, E7 targets pRb for degradation, promoting the transcriptional activation of S-phase genes by E2F. For a more exhaustive review of this topic, see [3,6,7,15,27,28,29,30,31,32,33].

## 3. Keratinocytes as Mediators of Innate Immunity Against HPV Infection

The innate immune response is the first line of defense against microbial pathogens. To detect and respond to the invading pathogens, cells utilize specialized receptor proteins, defined as pattern recognition receptors (PRRs), which bind to pathogen-associated molecular patterns (PAMPs), a set of viral proteins and nucleic acids, including double-stranded RNA, single-stranded RNA, CpG unmethylated DNA and 5′ triphosphorylated RNA (ppp-RNA) [34,35,36,37,38,39,40,41,42,43,44].

Based on their location, PRRs are classified as membrane-bound or unbound intracellular receptors. The former comprises Toll-like receptors (TLRs) and C-type lectin receptors (CLRs), which are located at the surface of cells or endocytic compartments. The latter includes leucine-rich repeat (LRR)-containing (or NOD-like) receptors (NLRs), RIG-I-like receptors (RLRs) and the AIM2-like receptors (ALRs), which are located in the cytoplasm to sense intracellular pathogens. Furthermore, a panel of structurally unrelated cytosolic DNA sensors has been identified, among which the most relevant is the enzyme cyclic guanosine monophosphate–adenosine monophosphate (cyclic GMP–AMP) synthase (cGAS), which in bound conformation with double-stranded DNA (dsDNA) catalyzes adenosine 5′-triphosphate (ATP) and guanosine 5′-triphosphate (GTP) into cyclic GMP–AMP (2′3′-cGAMP). This second messenger binds to and activates the stimulator of interferon genes (STING), a scaffold protein localized on the endoplasmic reticulum (ER) membrane. Although PRRs are different in terms of structure, the point of sensing and adaptor partner, they considerably overlap in their use of downstream signaling components and enzymatic pathways, ultimately leading to the activation of the transcription factors NF-κB and IRF3/7. While NF-κB triggers transcriptional activation of pro-inflammatory cytokines and chemoattractant cytokines and chemokines, IRF3/7 induces type I (IFNβ) and III (IFNλ) interferons (IFNs). IFNs are then key autocrine and paracrine regulators of the antiviral state thanks to their ability to induce a plethora of interferon-stimulated genes (ISGs) that can inhibit viral replication while triggering cell growth arrest and apoptosis [34,35,45,46,47,48,49,50,51,52,53,54,55,56,57,58,59,60,61,62,63,64,65]. 

The natural target cells of HPV are keratinocytes that form the stratified squamous epithelium of the skin and mucosal sites (e.g., the ano-genital or upper respiratory tract). Despite the fact that keratinocytes express several PRRs, which are able to sense viral pathogens and promote the innate immune response, HPVs have been very successful throughout evolution in creating an unreactive cellular milieu suitable for viral replication, persistence, and tumorigenesis [66,67,68,69,70,71,72,73,74,75,76,77,78,79,80,81,82].

The consensus model of cervical cancer progression clearly indicates as unifying risk factors the establishment of a persistent infection by hrHPV genotypes and the deregulation of normal viral gene expression, leading to steady-state E6/E7 overexpression. It is now widely accepted that HPV viral DNA integrates into the host genome quite frequently, especially during persistent HPV infection by hr genotypes. This event in turn stabilizes transcription of E6 and E7, thus conferring a growth advantage to the host cells. Despite the fact that approximately 90% of all HPV infections are resolved by the host immune system within a 2-year period, some patients may develop persistent HPV infections for years and eventually develop cancer, thus indicating that the innate immune reaction of the infected cells has been impaired in these cases. Even though the molecular determinants of this deficient immune response are potentially druggable targets to restore the innate immune response in HPV-transformed cells, they are still poorly characterized [2,3,6,8,15,16,17,76,83,84].

A major obstacle to understanding the true impact of E6/E7 on the host immune response is the difficulty in putting together pieces of data obtained using different experimental model and cell lines, which can easily create bias, especially when dealing with innate immunity. Indeed, current experimental models to study HPV-associated tumors are notoriously flawed by a number of limitations, mostly because HPV can only replicate and complete its life cycle in stratified squamous epithelial sites, which are quite difficult to model in vivo. Moreover, HPV can complete its viral life cycle in vitro only when keratinocytes are grown in organotypic raft culture, but, even in this case, the efficiency of viral replication is usually quite low. To further complicate matters, no animal model of infection is currently available for hr human genotypes (e.g., HPV16 and HPV18). Thus, the study of the cross talk between HPV and host cells has been mostly carried out in keratinocytes grown as monolayers. This cell system unfortunately has a series of caveats affecting the interpretation and generalizability of the findings, with the heterogeneity of the model of HPV infection being at the top of the list. In short, currently available cell models of HPV infection/transformation are based on the following strategies: (i) infection with HPV16 pseudovirions or virions; (ii) lentiviral-mediated overexpression of E6 and E7 from HPV16 or HPV18 by; (iii) transfection with episomal viral genomes; and (iv) transformed cells harboring multiple copies of the integrated or episomal viral genome.

With regard to HPV-associated disease models, it is particularly crucial to distinguish between models of infection where the viral genome is replicating in the infected cells and those where the pathogenic impact of the virus is solely due to deregulated E6/E7 expression. This latter model appears to fully recapitulate the status of HPV-transformed cells given that the viral genome is no longer replicating and, in most cases, is integrated as a partial genome. This condition can also be artificially obtained when cells are stably transduced with retroviral vectors expressing E6 and E7 or in proliferating cells stably maintaining episomal viral genomes after several passages in culture. The interface or interplay of the virus with the host innate immune system under these two different physical statuses of the virus, also characterized by a distinct pattern of viral protein expression, has not been fully elucidated yet. In addition, while a number of excellent reviews have already addressed the events occurring during the early phases of HPV infection, describing the cross talk between the host cell innate immune system and viral replication, there is lack of reviews focusing on the molecular mechanisms underlying the E6/E7-mediated dampening of the innate immune response in HPV-transformed keratinocytes, where these oncoproteins are aberrantly regulated [72,73,74,75,76,85]. Thus, here we take a close look at recent advances showing how viral immune escape mechanisms may contribute to the development and maintenance of a transformed phenotype during HPV-driven carcinogenesis. Since HPV16 and HPV18 account for almost 70% of all HPV-associated cancers, our analysis will be predominantly focused on the literature dealing with these two genotypes [1,16,17,18,86].

## 4. Impact of E6/E7 on the Innate Immune Response

A series of reports dating back to the first decade of the 2000s showed that hrHPVs inhibit several ISG transcripts, mainly through E6/E7. The fact that many of these targeted ISGs were involved in the antiviral response led us to hypothesize that the virus could establish a persistent state of infection conducive to cancer development by evading host immunosurveillance [71,87,88,89].

Important insights into the molecular mechanisms of viral immune evasion came from studies showing how E6/E7 proteins could bind and inactivate transcription factors, such as IRF1, IRF3, and STAT1, thereby influencing either the transcriptional activation of IFN genes or the downstream IFN receptor pathways that lead to the transcriptional activation of ISG genes [70,71,73,76,79,88,90,91,92,93]. These findings, together with a better understanding of the underlying mechanisms of the PRR-mediated induction of IFN, further supported the hypothesis that E6/E7 oncoproteins played a central role in manipulating the cellular environment to create an unreactive state suitable for uncontrolled cell proliferation.

The sections below summarize emerging evidence of the impact of E6/E7 viral proteins on the three major innate immune signaling pathways, namely cGAS/STING/TANK binding kinase-1 (TBK1), RIG-I/MAVS/TBK1, and TLRs, regulating the transcriptional activation of IFNs, pro-inflammatory cytokines, and chemokines in response to exogenous stimuli. We will mainly focus on nucleic acid recognition mechanisms and downstream signaling, highlighting the strategies devised by hrHPVs to escape from innate immunity and drive carcinogenesis. For quick reference, these data have also been listed in Table 1.

### 4.1. cGAS/STING/TBK1

Since the discovery of STING (also known as MPYS, MITA, or ERIS) as a powerful inducer of IFNs, several putative DNA sensors have been reported to survey the presence of cytosolic DNA and activate type I IFN production through STING signaling. However, it is now widely acknowledged that the major pathways mediating the immune response to exogenous DNA rely on the cGAS–STING axis, as outlined in Figure 1, and for this reason only this pathway will be considered in this review [40,41,42,43,44,62,105].

Upon DNA binding, cGAS converts adenosine 5′-triphosphate (ATP) and guanosine 5′-triphosphate (GTP) into cyclic GMP–AMP (2′3′-cGAMP), which functions as a secondary messenger that binds and activates STING. Its activation leads to trafficking from the endoplasmic reticulum (ER) to an ER-Golgi intermediate compartment followed by its migration into the Golgi apparatus and eventually into the perinuclear regions, where it is degraded. During trafficking, STING recruits and activates the kinase TBK1, which in turn phosphorylates IRF3, leading to IRF3-mediated IFN induction [58,60,61].

Even though cGAS–STING signaling is pivotal to withstanding infection by numerous pathogens, recent studies have unveiled additional functions of this pathway other than antimicrobial immunity. For instance, the cGAS–STING axis plays a role in activating a sterile cellular stress response, such as that occurring in cancer cells due to chromosomal abnormality, genomic DNA damage, and hyperproliferation. As cancer cells often form micronuclei or cytoplasmic DNA that may activate cGAS–STING signaling, it has become increasingly clear that the inactivation of this pathway is an efficient strategy for the virus to limit the induction of inflammatory and immune stimulatory molecules with tumor-killing activity (e.g., cytokines and NK ligands) [59,105,106,107,108,109,110,111,112,113]. Experimental results corroborating a model whereby HPV induces transformation and neoplastic progression also through the inhibition of the cGAS–STING immunomodulatory pathway are outlined below.

The first study showing the direct inhibition of cGAS by HPV oncoproteins was published in 2015 by Lau et al. This group transduced primary embryo fibroblasts (MEFs) with retroviruses expressing either HPV18 E6 or E7 and found that those transduced with E7, but not E6, lost their ability to secrete type I IFN in response to exogenous DNA stimulation. By contrast, the authors failed to observe any inhibition when RIG-I agonists were used to stimulate these cells. Of note, they could not demonstrate any direct interaction between E7 and cGAS, while they were able to co-immunoprecipitate STING and E7 by overexpressing a Flag-STING construct in HeLa cells, a cervical carcinoma-derived cell line harboring integrated HPV18 DNA. Consistent with an immune suppressing activity of HPV18, disruption of the E7 gene in HeLa cells through a lenti-CRISPR-Cas9 approach restored responsiveness to exogenous DNA in terms of IFN production [94].

These findings were later confirmed and further strengthened by a study from our group showing that keratinocytes stably harboring multiple copies of hrHPV18 genome (i.e., NIKSmcHPV18), as well as HeLa cells, displayed a marked transcriptional downregulation of the PRRs cGAS, its downstream adaptor STING, and, to a lower extent, RIG-I. Likewise, we observed reduced expression levels of the transcription factors IRF1 and 7. It is important to point out that, in our experiments, we exclusively used NIKSmcHPV18 cells grown between passages 20 and 30, when E6 and E7 transcripts are higher than that of E2, an expression pattern typical of persistent HPV infection. Overall, our findings indicate that the HPV-induced repression of PRR signaling is probably the reason why keratinocytes can maintain a high copy number of episomal viral DNA without triggering an antiviral response. Fittingly, we demonstrated that HPV18 persistence in keratinocytes inhibits both type I and III IFN production in response to DNA ligands, and that this effect is mainly due to suppression of the cGAS–STING pathway, which seems to be irreversible even after treatment with exogenous DNA. The downregulation of cGAS–STING signaling occurred through transcriptional repression of each single component via a novel epigenetic silencing mechanism, as attested by the accumulation of repressive heterochromatin markers, mainly H3K9me2, at the promoter region of RIG-I, cGAS, and STING genes [95]. We also showed that the accumulation of this repressive marker was due to the E7-dependent transcriptional induction of the H3K9-specific methyltransferase SUV39H1, the human homolog of the Drosophila Su(var)3-9 histone methyltransferase, which triggers histone H3Lys9 trimethylation (H3K9me3), inducing a chromatin conformational transition from an open to a closed state. In good agreement, the drug-induced inactivation or gene silencing of SUV39H1 disrupted H3K9me2/me3 binding to the promoter regions of RIG-I, cGAS, and STING, promoting the transcriptional activation of RIG-I and cGAS alongside an augmented secretion of type I and III IFNs. Importantly, SUV39H1-mediated chromatin remodeling in HPV18-harboring HeLa cells was similar to that observed in HPV16-harboring CaSki cells [114], implying that both hr genotypes have developed evolutionarily conserved strategies to epigenetically overturn the mechanisms of immune surveillance [96].

Another significant breakthrough to the field was later made by Luo et al. who provided the first evidence that the HPV-mediated inhibition of the STING signaling pathway may also play a significant role in the biology of head and neck squamous cell carcinoma (HNSCC). Through tissue microarray analysis of 297 HNSCCs (32% HPV^+^ and 60% HPV^−^), Luo and co-workers were able to demonstrate that high STING scores in the parenchyma and tumor microenvironment (TME) of these tumors correlated with improved patient survival. When the variables age, stage, site, HPV, and smoking habits were taken into account by multivariate Cox regression, they found that STING scores in the tumor parenchyma but not TME were still indicative of better survival rates, underscoring the prognostic value of STING expression in HNSCC patients. The same authors went on showing that STING expression levels, as well as those of TBK1 phosphorylation, were strongly reduced upon the exogenous expression of HPV16 E7 in a panel of HNSCC-derived cell lines. Consistently, these events were accompanied by reduced type I IFN induction upon poly(dA:dT) transfection, while E7 depletion in HPV16^+^ HNSCC cells restored IFN production following exogenous stimulation. Importantly, they found that HPV16 E7 but not HPV18 E7 hijacks NLRX1 to induce STING degradation via an autophagy-dependent mechanism, thereby providing a mechanism of STING downregulation in HNSCC cells where HPV16E7 does not bind STING [97].

Although first implicated in immune regulation, NLRX1 was later shown to regulate the mitochondrial pathway of apoptosis and IFN production via MAVS-RLR signaling. Specifically, NLRX1 attenuated type I IFN production while promoting autophagy during viral infection. NLRX1 is a component of a regulatory mitochondrial protein complex that also comprises the mitochondrial Tu translation elongation factor (TUFM), which, following viral infection, mediates the induction of IFN-I and autophagy through RIG-I and the autophagy-related proteins Atg5-Atg12 conjugate and Atg16L1 [115,116,117]. In this regard, Luo et al. found that NLRX1/E7 interaction contributed to enhanced STING turnover in HNSCC cells. Thanks to a newly characterized HPV16 E6/E7-expressing HNSCC mouse model (MOC2-E6/E7), syngeneic to C57BL/6, they also demonstrated that the CRISPR/Cas9-mediated depletion of NLRX1 in tumor cells could significantly increase STING activation upon poly(dA:dT) stimulation, as determined by the enhanced expression levels of TNF-α, IL-6 and IFNβ1, key markers of STING-mediated downstream effector activation. They also found that the depletion of NLRX1 in tumor cells significantly reduced the number of tumor-infiltrating lymphocytes in an IFN-dependent fashion accompanied by a more efficient cytotoxic T lymphocytes (CTLs) expansion in the draining lymph nodes [97].

Consistent with the epigenetic downregulation of STING gene expression in cancer cells hampering IFN and cytokine production upon cytosolic DNA stimulation, Qin Yan and collaborators demonstrated that STING mRNA expression is epigenetically downregulated by the histone H3K4 lysine demethylases KDM5B and KDM5C, whereas it is activated by H3K4 methyltransferases. In breast cancer cells, following KDM5 inhibition, STING expression was upregulated, thereby permitting interferon production in response to cytosolic DNA. Of note, a negative correlation between KDM5B and STING expression was similarly observed in HPV^+^ head and neck and cervical cancers. The expression of KDM5B was also negatively associated with CXCL10 expression—i.e., the interferon-inducible chemokine promoting infiltration of immune cells into the tumor microenvironment—and CD8+ T-cell infiltration [98].

In addition, the dampening of the cGAS–STING signaling pathways has also been reported in a panel of HNSCC-derived cell lines with the documented expression of HPV16 E7. Accordingly, in these cell lines, IFNβ induction upon stimulation with salmon sperm or 2′3′-cGAMP, two potent agonists of this pathway, was significantly impaired when compared to HPV^−^ HNSCC-derived cell lines. Conversely, no significant difference in terms of IFN induction upon 5′ppp-dsRNA stimulation was observed between HPV^+^ and HPV^−^ cell lines, indicating that the RIG-I signaling pathway is not impaired by HPV16 E7 expression, at least in the cell lines tested. One limitation of this study is that IFN induction was only measured at the mRNA level [99].

Collectively, the aforementioned seminal studies uncover a novel strategy of HPV16 immune evasion aimed at reducing the availability of the key innate immune sentinel STING. Since NLRX1 has also been reported to regulate the MAVS/RLR signaling pathway, one may envisage that impairment of RIG-I functions may also be a common feature of HPV16-associated cancers, a hypothesis that awaits further confirmation.

### 4.2. RIG-I/MAVS/TBK1

While various sensors play a role in recognizing non-self DNA when aberrantly expressed or localized within a cell, RLRs appear to be the sole receptors involved in the cytosolic recognition of viral RNA, with RIG-I being the founding member of the family. Although RIG-I was originally identified as a crucial cytoplasmic sensor surveying the presence of RNA viruses, mounting evidence indicates that it also plays a role in detecting several DNA viruses as well as RNA species generated by RNA polymerase III [118,119]. In addition, physical and functional interconnections between RIG-I and STING have been reported during infection with both RNA and DNA viruses, which are often associated with enhanced antiviral response. The role of STING in mediating RIG-I signaling is so crucial that many RNA viruses have learned how to block this signaling cascade to escape from immunosurveillance [41,120,121,122,123,124,125]. Whether this interconnection also exists in the case of HPV remains to be clarified.

RLRs, such as RIG-I and MDA5, harbor a central DEAD box helicase/ATPase domain and a C-terminal regulatory domain (CTD), required for RNA binding and, at least for RIG-I, the self-repression of RLR activity. As depicted in Figure 2, upon binding of 5′-ppp-RNA to the CTD/helicase region, RIG-I is regulated by a series of post-translational modifications (e.g., dephosphorylation and ubiquitination) and ATP-dependent conformational changes that promote its oligomerization and binding to the adaptor MAVS. Once activated, RIG-I/MAVS signaling bifurcates into two distinct molecular cascades: one pathway involves the TBK1 and IκB kinase ε (IKKε), which directly phosphorylate IRF3/7 to transcriptionally activate type I and type III IFNs; the other pathway engages the IKKα/β/γ complex, which promotes NF-κB-dependent upregulation of proinflammatory genes (Figure 2) [53,54,100,121,122,126].

Although to date there is no evidence of the involvement of RIG-I in HPV sensing, the inactivation of this signaling pathway through distinct mechanisms has been reported. In particular, Chiang et al. demonstrated that HPV16 E6 forms a complex with TRIM25 and its upstream regulator, the ubiquitin-specific protease 15 (USP15). As TRIM25 protein stability is regulated by the balance between degradative K48-linked ubiquitination and USP15-mediated deubiquitination, the authors performed coimmunoprecipitation experiments in HEK 293T and cervical-carcinoma-derived C33a cells ectopically expressing FLAG-tagged E6 of HPV16, showing that E6 binds exogenous TRIM25 and USP15, giving rise to a ternary E6-TRIM25-USP15 complex. Enhanced E6-driven TRIM25 polyubiquitination led to reduced TRIM25 protein-stability. It is worth noting that the ability of E6 to form a ternary complex was also observed in other hr and low-risk HPVs from the alpha genus. In contrast, E7 from the same genotypes failed to bind TRIM25, indicating specificity of this inhibitory cascade. Consistently, enhanced TRIM25 destabilization in HPV16E6 expressing cells hindered the interaction of RIG-I with its downstream adaptor mitochondrial antiviral signaling protein (MAVS) upon Sendai virus infection. As expected, the ectopic expression of HPV16 E6 but not E7 in these cells strongly inhibited the RIG-I-mediated induction of ISGs [100,126]. Although the authors provide compelling evidence that the ectopic expression of E6 can suppress TRIM25-dependent K63-linked ubiquitination of RIG-I and its caspase activation and recruitment domain (CARD) dependent interaction with MAVS, RIG-I ubiquitination status and activity in HPV-transformed cells, such as HeLa or CaSki, was not assessed. The latter would have allowed the authors to corroborate their findings in a cell environment where both E6 and E7 are expressed as it occurs in HPV-driven cancer. Considering that our group has recently reported the SUV39H1-dependent epigenetic silencing of RIG-I in HPV-transformed cells (i.e., HPV16, HPV18, and HPV31), it will be crucial to determine whether the E6-mediated post-translational modification of TRIM25 and E7-induced transcriptional repression of RIG-I are two coexisting and convergent pathways of HPV immune escape [95,96].

### 4.3. TLRs

TLRs are expressed on the cell surface and in endosomal compartments to respond to extracellular and endosomal PAMPs or danger associated molecular patterns (DAMPs), produced by cells in distress, such as that caused by viral infections. Cell surface TLRs mainly recognize microbial membrane components. In particular, TLR4 binds bacterial lipopolysaccharide (LPS), whereas TLR1, TLR2, and TLR6 recognize a large number of PAMPs, such as peptidoglycans, lipoproteins, and lipoteichoic acids. In addition, TLR5 specifically senses bacterial flagellin. In contrast, TLRs 3, 7, 8, and 9 generally localize to endosomal membranes, where they sense a wide range of nucleic acids derived from viruses as well as endogenous nucleic acids that exit the nuclei under stress conditions. As outlined in Figure 3, upon dsRNA binding, TLR3 regulates the activation of the transcription factor IRF3 that induces the expression of IFNs. TLR7 and TLR8 recognize ssRNA, whereas TLR9 binds DNA containing unmethylated cytosine-phosphate-guanosine (CpG) motifs. TLRs 7, 8, and 9 act through the myeloid differentiation primary response gene 88 (MyD88) pathway, which promotes the recruitment of downstream signaling molecules such as kinases that phosphorylate the inhibitor of kappa-B (IκB) to induce nuclear factor kappa-light chain-enhancer of activated B cells (NF-κB) signaling or IRF7 to induce the expression of type IFNs or proinflammatory cytokine genes, respectively [45,46,127]. 

Immortalized epithelial cells from the lower female reproductive tract, including End1/E6E7 were reported to express mRNA for all 10 human TLRs, with the exception of TLR4 [128]. Tommasino and co-workers reported the down regulation of TLR9 expression at both the mRNA and protein levels in HPV16 E6/E7-transduced keratinocytes. TLR9 suppression occurred at the transcriptional level as no TLR9 transcript was found in any of the three HPV16 E6/E7 keratinocyte lines generated independently using primary cells from different donors. Conversely, in three HPV18 E6/E7-transduced cell lines, the authors could not detect any significant variation in TLR9 mRNA levels when compared to normal keratinocytes. A similar trend was found in cervical carcinoma-derived cell lines. Specifically, SiHa and CaSki cells, containing an integrated HPV16 DNA, displayed respectively weak and undetectable levels of TLR9 mRNA, whereas HeLa cells, harboring an integrated HPV18, showed much higher TLR9 mRNA expression levels than those seen in SiHa, confirming the lower efficiency of HPV18 in inhibiting TLR9 transcription. Accordingly, TLR9 downregulation was also observed in cervical cancer biopsies by immunohistochemistry [101]. 

The inhibition of TLR expression was also observed in other studies assessing mRNA expression levels of TLR2, TLR3, TLR7, TLR8, and TLR9 genes in cervical cytobrush samples. In particular, it was found a significant association between enhanced TLR3 or TLR7 expression at an HPV16-positive visit and clearance by the following visit. These higher TLR expression levels were associated with increased levels of IFNα2, as judged by immunoassay in cervical lavage specimens. Accordingly, women with CIN2 regression had significantly higher baseline levels of TLR2, TLR7, and TLR8 when compared to women with CIN2 persistence/progression [102,103,104].

Collectively, these clinical data point to a critical role of TLRs in controlling HPV16 infection and suggest that the dampening of TLR expression in the cervical mucosa is a mechanism whereby HPV16 interferes with innate immune responses, contributing to viral persistence. However, despite the aforementioned clinical evidence, the underlying mechanisms of the HPV16-mediated downregulation of TLRs have yet to be fully characterized.

### 4.4. PRR-Related Transcription Factors

Lastly, it is worth considering two earlier studies showing for the first time the E6/E7-mediated inhibition of IRF1/3 transcriptional activity. Although this inhibition was initially thought to impair IFN activity, we now know that the same players are also involved in PRR signaling, indicating that HPV viral oncoproteins can tamper with downstream effectors of PRRs as well. In particular, HPV16 E6, but not HPV18 E6, was shown to bind and suppress the transcriptional activity of IRF3. More recently, Sangdun Choi and collaborators performed a series of in silico examinations, indicating that the LxxLL motifs of IRF3 binds within the hydrophobic pocket of E6, thus precluding Ser-patch phosphorylation, necessary for IRF3 activation and IFN induction. However, these data have yet to be confirmed experimentally. IRF1 is also a target of HPV16 E7 as its expression was reported to abrogate IRF1 DNA binding activity. A physical interaction between IRF1 and E7 that requires the carboxyl-terminal transactivation domain of IRF1 and the pRb-binding portion of E7 has also been reported [90,91,92,93,129].

## 5. Concluding Remarks and Future Perspectives

In conclusion, we have reviewed past and current literature addressing common strategies evolved by hrHPV genotypes to subvert the innate immune response in keratinocytes and create an unreactive cellular milieu conducive to cell transformation. The hallmark of HPV persistence—the conditio sine qua non of cancer development—is undoubtedly the deregulated expression of the two viral oncoproteins E6 and E7, whose most regarded function is that of interfering with the tumor suppressors p53 and pRb. By targeting cell cycle checkpoints and apoptosis pathways, E6 and E7 put host cells at increased risk for cellular genomic instability and chromosome abnormality, major driving forces of carcinogenesis. On the other hand, host cells are equipped with several stringent and intertwined signaling pathways that enable them to maintain genome integrity through the immune clearance of damaged cells, thereby preventing malignant transformation.

Even though DNA damage response has been largely implicated in the regulation of genome integrity and cell death, recent findings have established a relationship between genomic instability and inflammation. In particular, recent studies have provided mechanistic insights into how DNA damage elicits the production of type I IFNs and other immunoregulatory cytokines. These events are mainly triggered by the localization of DNA to the cytoplasm, mostly in the form of micronuclei, which, after being sensed by cytoplasmic PRRs, trigger downstream signaling cascades [59,108,130,131].

Consistent with cGAS–STING signaling playing a major role in the activation of IFNs and cytokines responsible for the clearance of potentially carcinogenic damaged/stressed cells, the frequent suppression of cGAS and STING expression or function has been reported in many types of human malignancies [105,106,107,108,131,132]. Accordingly, the impact of E6/E7 on innate immunity rather than cell proliferation has gained increasing attention as one of the major determinants of HPV-induced carcinogenesis. Notably, the observed inhibition of cGAS–STING signaling in HPV-transformed cells may also provide an explanation as to why cells harboring hrHPV infection are still capable of replicating despite the activation of the DNA damage response, which would otherwise arrest cellular replication by triggering the innate immune response [108,131,133,134].

While the immune evasion strategies evolved by HPV to establish viral latency have already been extensively addressed, we have summarized a growing body of literature echoing that the dampening of the innate immune response by E6/E7 plays an equally important role in tumor development. Another important take-home message is that HPV, thanks to the concerted actions of E6 and E7, has evolved multi-faceted mechanisms aimed to curb multiple rather than single downstream effectors of the immunosurveillance network. In particular, E7 appears to be mostly responsible for PRR suppression. The nature of this inhibition, including the reversibility of the modifications determined by E7 expression, seems to be quite different, especially when we consider the cGAS–STING and RIG-I signaling pathways [94,95,96,97,98,99,100]. Indeed, while the former is heavily impaired in HPV-transformed cells, and thus hardly reactivatable, RIG-I signaling is still partially functional, and when stimulated by powerful agonist, e.g., M8, it can boost a relatively large amount of IFNs, especially type III IFN [95,96]. Therefore, the specific targeting of immune pathways such as RIG-I appears to be an attractive and feasible option to trigger the innate immune response in HPV-transformed cells. Considering the emerging role of M8 as inducer of immunogenic cell death in cancer cells through the production of type I IFNs and lymphocyte-recruiting chemokines, this compound may be a suitable candidate to enhance cell death in HPV-transformed cells and improve the effectiveness of existing anticancer therapies in HPV-associated tumors in vivo [135,136,137,138,139].

It is our hope that the findings and ideas discussed herein will spur cross-disciplinary investigations aimed at clarifying the functional role of viral oncoproteins at the intersection of immune evasion and aberrant proliferation in HPV-associated cancers, with the long-range goal to discover novel targets for therapeutic development.

## Figures and Tables

**Figure 1 pathogens-09-00292-f001:**
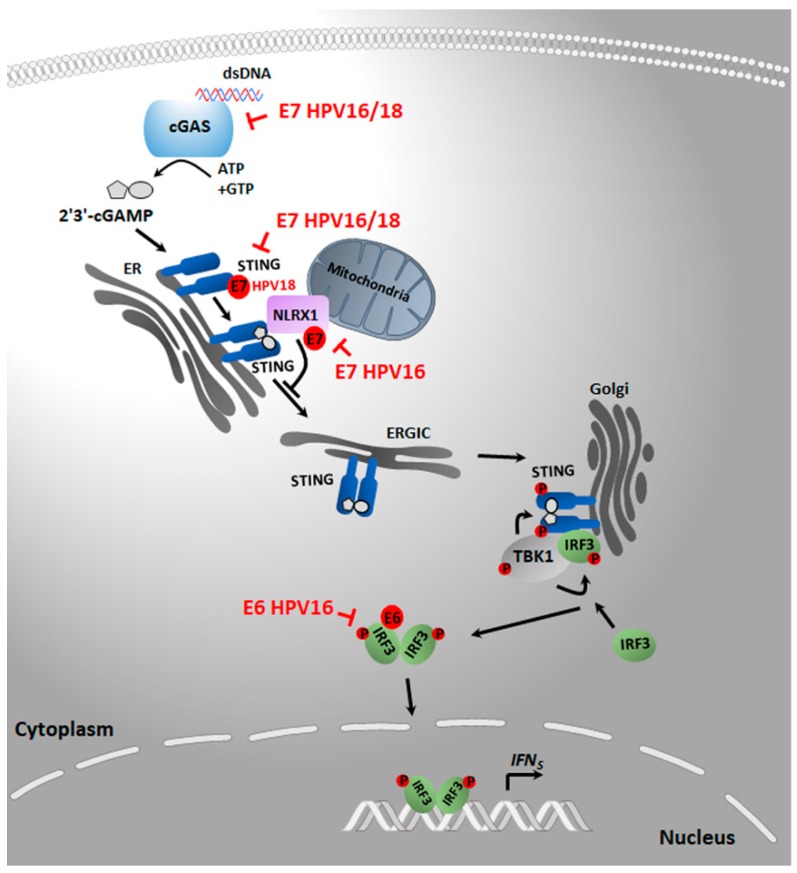
DNA sensing mechanisms through the cGAS/STING/TBK1 signaling pathway. Upon DNA binding, cGAS undergoes a conformational change that leads to its activation. Activated cGAS catalyzes the synthesis of the second messenger cyclic GMP–AMP (2′3′-cGAMP) from ATP and GTP, which is a STING ligand. 2′3′-cGAMP binding to STING results in the translocation of STING from the ER compartment to the ER–Golgi intermediate compartment (ERGIC) and the Golgi apparatus, triggering the activation of the following downstream signaling pathways: STING recruits autophosphorylated TBK1, which then phosphorylates STING, thereby promoting the docking of the transcription factor IRF3 to the phosphorylated STING residue, resulting in the TBK1-dependent phosphorylation of IRF3. Phosphorylated IRF3 dimerizes and translocates to the nucleus, where it transcriptionally activates IFNs genes. The binding between E7 and the mitochondrial NLRX1, which in turns contributes to enhance STING turnover, is also indicated. The symbol ┫ indicates the specific target of the inhibitory action of E6 or E7 oncoprotein. When a direct binding of the oncoprotein to a specific target has been demonstrated, this is indicated by the symbol ●. Abbreviations: cGAS, cyclic GMP–AMP synthase; dsDNA, double-stranded DNA; ATP, adenosine triphosphate; GTP, guanosine triphosphate; 2′3′-cGAMP, cyclic GMP–AMP; ER, endoplasmic reticulum; STING, stimulator of interferon genes; NLRX1 Nucleotide-Binding Oligomerization Domain, Leucine Rich Repeat Containing X1; ERGIC, endoplasmic reticulum-Golgi intermediate compartment; TBK1, TANK-binding kinase 1; IRF3, interferon regulatory factor 3; P, phosphorylation; IFN, interferon; HPV, human papillomavirus.

**Figure 2 pathogens-09-00292-f002:**
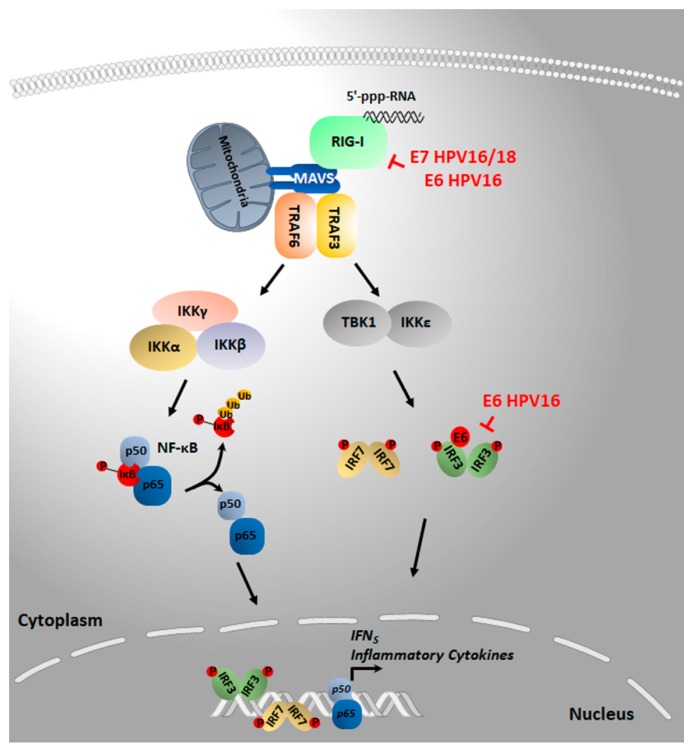
Upon 5′-ppp-RNA binding, RIG-I engages the adaptor protein MAVS on the mitochondrial outer membrane. MAVS activation mediates the assembly of a signaling complex comprising TRAF 3 and 6. MAVS signaling in turn activates TBK1, IKKε, and the IKKα/β/γ complex, triggering IRF3/7 and NF-κB activation, respectively. Upon translocation to the nucleus, IRF3/7 and NF-κB induce the transcription of IFNs and proinflammatory cytokines. The symbol ┫ indicates the specific target of the inhibitory action of E6 or E7 oncoprotein. When a direct binding of the oncoprotein to a specific target has been demonstrated, this is indicated by the symbol ●. Abbreviations: RIG-I, retinoic acid-inducible gene I; MAVS, adaptor mitochondrial antiviral signaling protein; TRAF, TNF receptor-associated factor; IKK, IκB kinase; TBK1, TANK-binding kinase 1; IRF, interferon regulatory factor; IκB, inhibitor of kappa B; NF-κB, nuclear factor-κB; Ub, ubiquitination; P, phosphorylation; IFN, interferon; HPV, human papillomavirus.

**Figure 3 pathogens-09-00292-f003:**
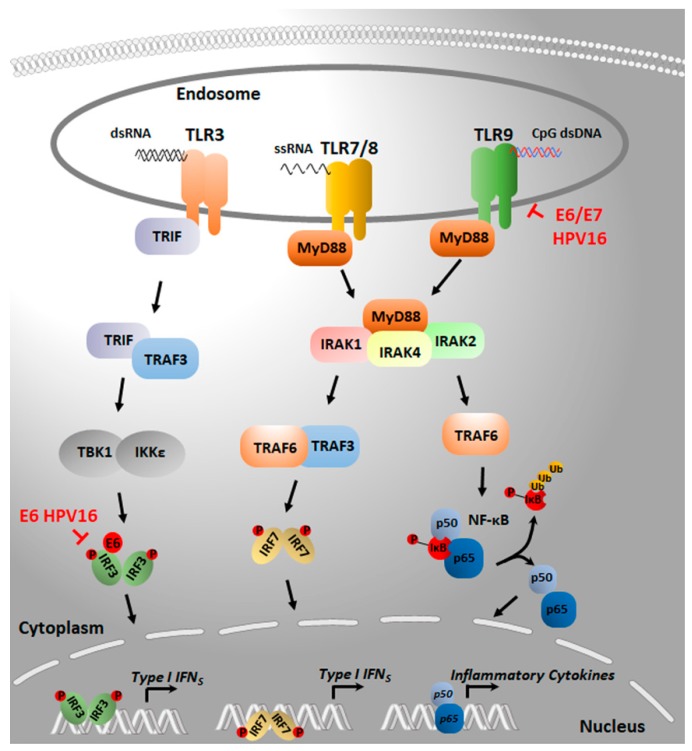
The endosomal nucleic acid sensing pathways. TLRs 3, 7, 8, and 9 typically localize to endosomal membranes, where they detect and bind to a variety of nucleic acids. Binding to their specific nucleic acid ligands leads to the formation of TLR dimers and the oligomerization of their cytoplasmic TIR domains, which recruit signaling adaptors as follows: TLR3 assembles with TRIF and then TRAF3 to activate the TBK1/IKKε/IRF3 axis for transcriptional activation of type I IFNs; TLRs 7, 8, and 9 form a complex with MyD88, promoting the formation of the Myddosome complex, which contains MyD88, IRAK4, IRAK1, and IRAK2. The formation of the Myddosome activates IRAKs and the ubiquitin E3 ligase TRAF6, which in turn promotes the NF-κB- and IRF7-mediated transcriptional activation of type I IFNs and pro-inflammatory cytokines. The symbol ┫ indicates the specific target of the inhibitory action of E6 or E7 oncoproteins. When a direct binding of the oncoprotein to a specific target has been demonstrated, this is indicated by the symbol ●. Abbreviations: TLR, Toll-like receptor; dsRNA, double-stranded RNA; ssRNA, single-stranded RNA; dsDNA, double-stranded DNA; TRIF, TIR-domain-containing adapter-inducing interferon-β; TRAF, TNF receptor-associated factor; IKK, IκB kinase; TBK1, TANK-binding kinase 1; MyD88, Myeloid differentiation primary response 88; IRAK, interleukin-1 receptor-associated kinase; IRF, interferon regulatory factor; NF-κB, nuclear factor-κB; IκB, inhibitor of kappa B; Ub, ubiquitination; P, phosphorylation; IFN, interferon; HPV, human papillomavirus.

**Table 1 pathogens-09-00292-t001:** Impact of E6/E7 oncoproteins from HPV16/18 on innate immunity signaling pathways.

Mechanism or Target	Cellular Model	Reference
**cGAS/STING/TBK1 signaling pathway**		
HPV18 E7 binds and antagonizes STING.	HeLa cells and mouse embryo fibroblasts stably transduced with HPV18 E6 or E7 expressing retroviruses.	Lau, L. et al., 2015 [94]
Epigenetic silencing of *cGAS* and *STING* genes through HPV16/18 E7-mediated induction of the methyltransferase SUV39H1.	HeLa, CaSki, NIKSmcHPV18 and HEK 293 cells expressing either HPV16 or HPV18 E6 and E7.	Albertini, S. et al., 2018 [95]; Lo Cigno, I. et al., 2020 [96]
HPV16 E7 hijacks NLRX1 to induce STING degradation via an autophagy-dependent mechanism.	HNSCC-derived cell lines ectopically expressing HPV16 E7 and a syngeneic C57/BL/6 model of HPV^+^ HNSCC.	Luo, X. et al., 2019 [97]
The H3K4 lysine demethylases KDM5B and KDM5C epigenetically suppress *STING* mRNA expression levels.	Breast cancer cells, HPV^+^ head and neck and cervical carcinomas.	Wu, L. et al., 2018 [98]
Impaired *IFNβ* gene transcriptional activation upon stimulation with STING agonists.	HPV16^+^ HNSCC-derived cell lines.	Shaikh, M.H. et al., 2019 [99]
**RIG-I/MAVS/TBK1 signaling pathway**		
HPV16 E6 forms a ternary E6-TRIM25-USP15 complex that reduces TRIM25 protein-stability, leading to reduced ubiquitination of RIG-I and suppression of its ability to interact with MAVS.	HEK 293T and the cervical carcinoma-derived cell line C33a ectopically expressing FLAG-tagged E6 of HPV16.	Chiang, C. et al., 2018 [100]
HPV16/18 E7 induces the transcription of SUV39H1, which promotes epigenetic silencing of RIG-I.	HeLa, CaSki, NIKSmcHPV18 and HEK 293 cells expressing either HPV16/18 E6 or E7.	Albertini, S. et al., 2018 [95]; Lo Cigno, I. et al., 2020 [96]
**TLR signaling pathway**		
HPV16 E6/E7 induce downregulation of TLR9 expression at both mRNA and protein levels.	Human primary keratinocytes stably transduced with HPV16E6/E7 expressing retroviruses, HeLa, SiHa and CaSki.	Hasan, U.A. et al., 2007 [101]
HPV16 suppresses TLR expression in the cervical mucosa, contributing to viral persistence.	Cervical cytobrush samples.	Daud, I.I. et al., 2011 [102]; Scott, M.E. et al., 2015 [103]; Halec, G. et al., 2018 [104]
**PRR-Related Transcription Factors**		
HPV16 E6 binds IRF3 and impairs its transcriptional activity.	In vitro synthesized protein, and HPV16 E6-transfected cells, including primary human keratinocytes.	Ronco, L.V. et al., 1998 [93]
HPV16 E7 binds IRF1 and impairs its DNA binding and transcriptional activity.	In vitro synthesized HPV16 E7 and HPV16E7-transfected cells.	Park, J.S. et al., 2000 [90]; Perea, S.E. et al., 2000 [91]; Um, S.J. et al., 2002 [92]

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
