# Peer review of "Subversion of Host Innate Immunity by Human Papillomavirus Oncoproteins"

_pathogens, 2020, doi:10.3390/pathogens9040292_

Round 1

Reviewer 1 Report

General comments:

This review is extremely well-written and very thoughtful- it is not merely a collection of literature results but a valuable assessment of an extremely complex field. I recommend immediate publication with only minor adjustments to the text and citations that the authors can address in the proofs.

Minor text issues:

Page 2, line 79. I object to the phrase, “it is highly likely that HPV-associated tumors will remain a major health concern for the near future.” Instead, it is highly likely that such tumors, and the need for therapy, will remain a concern for the foreseeable future, especially since men are carriers (and HPV cancer victims), and even a smaller percentage of men are vaccinated than that of women.

P 3, l 106. Grammar- “a pattern,…,has been recognized” (not “…have been recognized”)

P3, line 132, “some women” should read “some patients”- women are the vast majority, but not the only, HPV cancer victims (especially for head-and-neck cancers), and persistent genital warts don’t discriminate between men and women, to my knowledge.  The sentence refers to persistent HPV infections without other qualifications or restrictions.

P3, l 143, “are grown as organotypic raft culture” should read “are grown in organotypic raft culture.”

P4, l 147, monolayers, not monolayer.

P4, l151-3, I do not think that this list of models accounts for cells such as W12E,1 which are a clinical isolate of virally-transformed cells from a precancerous lesion that remain primary keratinocytes but carry HPV16 episomes.  W12E cells were not “artificially transfected with episomes.” W12E cells have also been used to generate organotypic rafts (yes, there are many caveats with such studies).

Comments about literature citations:

In the introduction, it would be helpful to cite a paper2 that describes the regression of tumors upon re-expression of p53 in tumor cells as caused by drugs such as chloroquine (a derivative of which was reported to eliminate warts in an unblinded clinical study3).

Also in the introduction, it would be valuable to cite a review of the role of the DNA damage response in HPV infections such as that cited here.4 Furthermore, some better-characterized, druggable targets have been found and accessed in cell and organotypic raft cultures.5-8

[1] Kim, K., Garner-Hamrick, P. A., Fisher, C., Lee, D., and Lambert, P. F. (2003) Methylation Patterns of Papillomaviral DNA, Its Influence on E2 Function and Implications in Viral Infection, J. Virol. 77, 12450-12459.

[2] Brown, C. J., Lain, S., Verma, C. S., Fersht, A. R., and Lane, D. P. (2009) Awakening guardian angels: drugging the p53 pathway, Nat Rev Cancer 9, 862-873.

[3] Bhushan, P., Aggarwal, A., and Baliyan, V. (2014) Complete clearance of cutaneous warts with hydroxychloroquine: Antiviral action?, Indian J. Dermatol. 59, 211.

[4] Fisher, C. (2015) Recent Insights into the Control of Human Papillomavirus (HPV) Genome Stability, Loss, and Degradation, J. Clin. Med. 4, 204-230.

[5] Castaneda, C. H., Scuderi, M. J., Edwards, T. G., G. Davis Harris, J., He, G., Dupureur, C. M., Koeller, K. J., Fisher, C., and Bashkin, J. K. (2016) Improved Antiviral Activity of a Polyamide Against High-Risk Human Papillomavirus Via N-Terminal Guanidinium Substitution, MedChemComm 7, 2076-2082.

[6] Edwards, T. G., Vidmar, T. J., Koeller, K., Bashkin, J. K., and Fisher, C. (2013) DNA Damage Repair Genes Controlling Human Papillomavirus (HPV) Episome Levels under Conditions of Stability and Extreme Instability, PLOS One 8, e75406.

[7] Edwards, T. G., Helmus, M. J., Koeller, K., Bashkin, J. K., and Fisher, C. (2013) Human papillomavirus episome stability is reduced by aphidicolin and controlled by DNA damage response pathways, J Virol 87, 3979-3989.

[8] Edwards, T. G., Koeller, K. J., Slomczynska, U., Fok, K., Helmus, M., Bashkin, J. K., and Fisher, C. (2011) HPV Episome Levels are Potently Decreased by Pyrrole-Imidazole Polyamides, Antiviral Research 91, 177-186.

Reviewer 2 Report

In this manuscript, Cigno et al. give an overview of recent studies bridging E6/E7-mediated inhibition of innate antiviral immunity and cancer progression. They first discussed the human carcinogenesis driven by HPV, revealing the essential role of E6 and E7 oncoproteins during that process. Then the authors focused on the pathogenic impact of the virus that is solely due to deregulated E6/E7 expression and reviewed the recent advances in exploring the molecular mechanisms underlying E6/E7-mediated dampening of the innate immune response in HPV-transformed cells. By summarizing the impact of E6/E7 on the innate immune signaling pathways, including cGAS/STING/TBK1, RIG-I/MAVS/TBK1 and TLRs, they demonstrate how viral immune escape mechanisms may contribute to the development and maintenance of a transformed phenotype during HPV-driven carcinogenesis. The review is comprehensive and informative. Here are some suggestions:

  1. As an important part of innate immunity, the downstream IFN receptor pathway is also targeted by E6/E7. It will be more comprehensive if this part could be discussed as well.
  2. The functional sensors in RLR pathway include RIG-I and MDA5. It is better to clarify when talking about it. For example, in line 343-344, “RLRs are characterized by a central DEAD box helicase/ATPase domain and a C-terminal 343 regulatory domain (CTD), essential for RNA binding and autorepression of RLR activity.” The CTD of RIG-I has the autorepression activity whereas CTD of MDA5 does not. Also, it will be more comprehensive to include MDA5 in the discussion.
  3. There is no reference cited in line 337. There are several research advances in RLR sensing during DNA virus infection, for example, PMID29180807, 29970461 and 

Reviewer 3 Report

The manuscript by Lo Cigno et al provides a in depth review of HPV immunoevasion tactics. The review centers around the viral oncogenes E6/E7 and their effect on the innate immune signaling pathways cGAS/STING/TBK1, RIG-I/MAVS/TBK1 and TLRs.  The review is well written and organized. This paper will be of interest to individuals studying viral oncogenesis and immune evasion.

I have no suggestions for improvement.

Author Response

Response to Reviewer 3 Comments

We thank the reviewer 3 for his/her comments and we would like to express our gratitude for his/her positive feedback.